# Citrus Taste Modification Potentials by Genetic Engineering

**DOI:** 10.3390/ijms20246194

**Published:** 2019-12-08

**Authors:** Li-Jun Li, Wan-Seng Tan, Wen-Jing Li, Yan-Bing Zhu, Yi-Sheng Cheng, Hui Ni

**Affiliations:** 1College of Food and Biological Engineering, Jimei University, Xiamen 361021, China; ljli@jmu.edu.cn (L.-J.L.); wansentan@jmu.edu.cn (W.-S.T.); yanbingzhu@jmu.edu.cn (Y.-B.Z.); 2Fujian Provincial Key Laboratory of Food Microbiology and Enzyme Engineering, Xiamen 361021, China; 3Research Center of Food Biotechnology of Xiamen City, Xiamen 361021, China; 4Department of Life Science, National Taiwan University, Taipei 10617, Taiwan; chengys@ntu.edu.tw; 5Institute of Plant Biology, National Taiwan University, Taipei 10617, Taiwan

**Keywords:** Citrus, taste, genetic modification, sour, bitter, sweet, metabolism

## Abstract

Citrus fruits are mainly consumed as fresh fruit and processed juice products. They serve as nutritional and a tasty diet in our daily life. However, the formidable bitterness and delayed bitterness significantly impact the citrus industry attributable to the two major bitter compounds naringin and limonin. The extremely sour and acidic also negatively affects the sensory quality of citrus products. Citrus breeding programs have developed different strategies to improve citrus quality and a wealth of studies have aimed to uncover the genetic and biochemical basis of citrus flavor. In this minireview, we outline the major genes characterized to be involved in pathways shaping the sweet, bitter, or sour taste in citrus, and discuss briefly about the possible approaches to modify citrus taste by genetic engineering.

## 1. Introduction

It was suggested that the *Citrus* genus originated from the Malay Archipelago and Southeast Asia, and the varieties of edible citrus fruits on the market are generated by hybridizations of ancestral species, natural or artificial mutations, and human selection during domestication [1,2]. The genealogy and genetic origin of the major citrus species are roughly summarized in Figure 1. The major ancestral species largely include mandarin (*Citrus reticulata*), pomelo (*Citrus maxima*), and citron (*Citrus medica*). The classification of commercial mandarins is very complicated, which largely include tangerine, satsuma, and clementine. Citrus hybrids are more diversified, and their taxonomy is somewhat inconsistent. The major hybrids as our common consumption include sweet orange (*Citrus sinensis*), bitter orange (*Citrus aurantim*), grapefruit (*Citrus paradisi*), lemon (*Citrus limon*), and limes [3]. Orange is a hybrid between pomelo and mandarin, and grapefruit is the product of a natural backcross of a sweet orange with a pomelo [4]. The original lemon is a hybrid of bitter orange and citron, while lime is highly diverse including several species. According to the Food and Agriculture Organization of the United Nations (FAO) citrus fruits statistics 2017 (FAO statistics, 2017), citrus fruit accounts for about 17% of total fruit production, in which about 50% and 23% of citrus fruits are orange and mandarins, respectively.

Citrus fruits have great nutritional and pharmacological values. Citrus is rich in vitamins, especially in vitamin C and is a good source of minerals and dietary fibers [5]. The phytochemicals such as monoterpenes, limonoids (triterpenes), flavonoids, carotenoids, and hydroxycinnamic acid have great potentials for the protection of humans against chronic diseases and cancer [6]. The citrus products consumed consist mostly of fresh fruits and fruit juice. The orange is more favored as processed orange juice, such as single-strength orange juice and frozen juice concentrate, which can be processed through up to 95% of orange fruit produced in Florida, US and 70% in Brazil, according the United States Department of Agriculture (USDA) report about citrus world markets released in July 2019. Generally, flavor is sensed by a combination of the taste and olfactory systems. Five primary taste sensations in our mouth are salty, sour, sweet, bitter, and umami [7]. As for citrus fruits and juice, they normally impress us with their sweet and sour. Although flavor is favored differently for different individuals, there is a generally accepted notion of good taste which is attributed to an appropriate balance of sweetness and sourness in citrus [7]. Excessive bitterness and the delayed bitterness in citrus pose great challenge for the citrus industry to provide consumers readily acceptable fresh fruit and citrus juice, thus causing great lose in citrus market value [8]. Extremely sour and acidic can destroy the sweet taste of citrus and may be physiologically harmful to humans. To improve citrus quality, traditional breeding techniques including cross-pollination, self-pollination, selection of natural mutations or induced mutations, and rootstock hybrid. Extensive studies have been performed to understand the citrus flavor-contributing factors through biochemical and genetic approaches [7,9]. In this review, we mainly focus on the understanding of molecular basis governing citrus sweetness, bitterness, and sourness, providing an overview of the potentials for genetic modification or molecular-assisted breeding to improve citrus overall taste.

## 2. Citrus Taste

### 2.1. Sweet

The content of three major carbohydrates (sucrose, fructose, and glucose) as the main components of total soluble solids (TTS) makes the major contribution to citrus sweetness [10]. Sucrose is exported to the developing citrus fruit from the source leaves with sugar accumulation through photosynthesis and conversion to translocation sucrose [11,12]. The partitioning of sucrose into citrus fruit was believed to be determined by the sink strength which is monitored by the enzyme activities of three sucrose-metabolizing enzymes, sucrose phosphate synthase (SPS), sucrose synthase (SS), and invertase (IVR) [13]. The schematic presentation of sucrose transport is shown in Figure 2. The sucrose accumulated in the sink leaves is loaded into the phloem through the well-recognized action of plasmalemma-localized sucrose/H^+^ symporter and transported to the filial storage site [14,15]. Sucrose is unloaded to the parenchyma tissue and further imported into citrus flesh through apoplastic and symplastic pathways. It was suggested that there existed P-ATPase involved transport of sucrose from apoplast to cytosol and an endocytic pathway of assimilating sucrose directly from apoplast into vacuole, while the sucrose/H^+^ antiporters for sucrose import into vacuole from cytosol remain to be identified if any [16]. The translocated sucrose is to be either hydrolyzed to fructose and glucose by IVR or cleaved to fructose and uridine diphosphate glucose (UDP-glucose) by SS, and then resynthesized by the activity of SPS. The two steps of sucrose degradation and resynthesis favor the sucrose import by generating a sucrose concentration gradient in the sink local zone for phloem unloading [17]. A collection of genes encoding the enzymes for sucrose transport and metabolism have been reported, including three sucrose transporters (CitSUT1, CitSUT2, and CitSUT3) for sucrose loading or unloading [18], three sucrose synthase isoforms (CitSUS1/2 and CitSUSA) from *Citrus unshiu* [17], and six SS homologs from publicly available genome database for sweet orange and clementine (citrus.hzau.edu.cn/orange and www.phytozome.net) [19], and their transcriptional and translational expression patterns were investigated during citrus fruit development. Generally, the sucrose synthase activity and expression levels are relatively low at the citrus fruit immature stage favoring sucrose accumulation and increased during fruit development favoring improving sink strength and sucrose import, while different sucrose synthases showed their unique expression patterns [19,20]. Komatsu et al. (2002) investigated the transcript levels of *CitSUS1/2* and *CitSUSA* during citrus development and suggested that CitSUS1 may help provide the sucrose degradation products for growth and cell wall construction while CitSUSA plays more roles in providing substrates for sucrose resynthesis in concert with the function of SPS [17]. Katz et al. (2012) performed proteomic and metabolic analysis during citrus fruit development and revealed that sucrose invertase expression largely remained unchanged while an invertase inhibitor was upregulated in the later stages of fruit development [21]. This further supported the notion that sucrose synthase is the major player mediating sucrose degradation and provides substrates for sucrose re-synthesis by SPS, as manifested that SPS showed co-upregulation with SS at later stages in the sink tissue.

Taken together, sucrose accumulation in citrus fruit is regulated at multiple levels during fruit development. A set of factors may determine the partitioning of sucrose into the fruits, including photosynthesis and conversion of translocation sucrose in leaves, sucrose loading into and unloading from phloem, and the coordination of the major sucrose metabolism-related enzymes and transporters [11]. It has been shown that drought stress or treatment can improve the sink strength by increasing the sucrose synthase activity thus enhancing sucrose importing into citrus fruit [18]. Given the involvement of a set of genes and knowledge of their spatial and temporal expression patterns, it is still not clear if a single gene modification would improve sucrose accumulation in citrus fruit until a comprehensive analysis of knockout mutants or controlled downregulation or upregulation of individual genes to examine their effects on sucrose partitioning.

### 2.2. Bitter

There are two types of bitterness, namely immediate bitterness and delayed bitterness, in citrus fruits, imparted by two different types of compounds [22,23]. The immediate bitterness is largely conferred by naringin and neohesperidin [1], and the delayed bitterness is mainly produced by limonin of limonoids [24]. Delayed bitterness is gradually developed upon fruit is mechanically damaged, juiced, or frozen [25,26]. Some enzymes are natural debittering enzymes providing citrus palatable quality. The overall picture of citrus bitter-tasting and non-bitter compounds synthesis pathways is sketched in Figure 3.

#### 2.2.1. Cm1,2RhaT

Flavanones are the major group of flavonoids in citrus and the primary bitterness in citrus fruit is determined by flavanone-neohesperidosides in bitter citrus species such as pummelo/pomelo, grapefruit, and bitter orange [27]. The flavanone naringenin is the dominant flavonoid backbone in some citrus species, while it might undergo a variety of modifications to form other flavanones, for instance, hydroxylation and methylation to generate its derivative hesperetin [1,28]. The bitter-tasting naringin is the major contributor to the bitter taste in grapefruit and neohesperidin in sour orange [27,29]. Naringinase is widely used as a debittering enzyme in the commercial production of citrus juice, and our lab has reported a mutated α-L-rhamnosidase of naringinase complex with enhanced performance on thermostability and citrus juice debittering efficacy [30].

In grapefruit, the bitter-tasting naringin is predominant over the tasteless narirutin. They are derived from the same flavanone skeleton naringenin but differ in the rhamnose-glucose disaccharide connected through an O-linkage at position 7. In naringin, the rhamnose is attached to the glucose moiety at the C-2 position, while it is attached at the C-6 position in narirutin, resulting in the formation of naringenin-7-O-neohesperidoside and naringenin-7-O-rutinoside, respectively. The gene for Cm1,2RhaT (flavanone 7-O-glucoside-1,2-rhamnosyltytransferase) was isolated from pummelo (*C. maxima*) [27], and its expression product of recombinant Cm1,2RhaT enzyme in tobacco BY2 cells exhibited UDP-glycosyltransferase activity specifically converting naringenin-7-O-glucoside into naringin. In addition, Cm1,2RhaT displayed similar enzymatic function on another flavonoid subgroup, flavone. Spatial and temporal expression patterns suggested that *Cm1,2RhaT* gene is highly expressed in young leaves and young fruit but is not actively transcribed in mature leaves and mature fruit, which match with the developmental accumulation pattern of flavanone-neohesperidosides [27,31]. *Cm1,2RhaT* expresses exclusively in naringin-containing citrus species (pummelo, grapefruit, and sour-orange), but not in sweet species like orange, citron, and mandarin [27]. A Cm1,2RhaT ortholog was identified with two frameshift mutations abolishing the functional 1,2RhaT expression in sweet orange (*C. sinensis*) [1]. These findings provide a possibility to modulate the expression of 1,2RhaT by genetic engineering to modify the bitter-tasting naringin content in those citrus species with readily detecTable 1,2RhaT expression.

#### 2.2.2. Cs1,6RhaT

In citrus, naringenin produced from naringenin chalcone catalyzed by chalcone isomerase can be further processed to form different subgroups of flavonoids featuring different structures of C ring, including flavanone, flavone, flavonol, and anthocyanidin [32]. After cloning and characterization of *Cm1,2RhaT* [27], Frydman et al. (2013) isolated *Cs1,6RhaT* (flavanone-7-O-glucoside-1,6- rhamnosyltytransferase) from orange (*C. sinensis*) which diverts another branch of glycosylation of flavanone-7-O-glucoside [1]. Cs1,6RhaT adds rhamnose to the C-6 position of glucoside leading to the formation of the tasteless flavanone-7-O-rutinoside, such as hesperidin and narirutin. Additionally, it turned out that Cs1,6RhaT is substrate-promiscuous which can catalyze 1,6-rhamnosylation of flavonoid-7-O-glucoside in flavanone, flavone, flavonol, and anthocyanidin as well as flavonoid-3-O-glucoside in flavonol and anthocyanidin. *Cs1,6RhaT* is highly expressed in the non-bitter citrus species such as orange, mandarin and lemon, but with relatively low expression in citron containing overall low levels of flavanones [33]. Contrary to *Cm1,2RhaT*, *Cs1,6RhaT* has low levels in grapefruit and bitter orange, and was not detected in pummelo which is rich in flavanone neohesperidosides but lacks rutinosides [1].

A more recent study by Ohashi et al. (2016) was performed on the branch-forming RhaT (rhamnose transferase) in the engineered fission yeast expressing Arabidopsis rhamnose synthase making UDP-Rha available in yeast system [34]. The *Cm1,6RhaT* was newly isolated from pummelo. Substrate preference studies of Cm1,2RhaT, Cs1,6RhaT, and Cm1,6RhaT indicated that Cm1,6RhaT can transfer rhamnose residue of UDP-rhamnose toward naringenin-7-O-gulcoside as Cs1,6RhaT; Cm1,2Rhat has glycosyl transfer activity using UDP-xylose in addition to UDP-rhamnose as donor toward naringenin-7-O-glucoside. Additionally, Cs1,6RhaT can transfer another rhamnose residue toward narirutin to yield naringenin-7-O-rhamnosylrutinoside which seems not to exist naturally in citrus species or is quite low in abundance if any [34]. Given that the amino acid sequences of Cm1,6RhaT and Cs1,6RhaT share about 94% in identity, the failure of the detection of Cm1,6RhaT by western blot using anti-Cs1,6RhaT antibody might be on account of the extremely low expression of Cm1,6RhaT in pummelo [1,34].

#### 2.2.3. CitdGlcT

1,2RhaT and 1,6RhaT confer the branched rhamnosylation of flavanone/flavone-7-O-glucoside in citrus. To further explore other enzymes governing the flavor-related glycosylation in citrus, especially the di-glycosylation, Chen et al. (2019) performed Basic Local Alignment Search Tool (BLAST) against citrus genome database (http://citrus.hzau.edu.cn/orange/ and phytozome.jgi.doe.gov) using *Cm1,2RhaT* CDS as query and identified another *1,2RhaT* and three *dGlcT* (di-glucosyltransferase) genes from citrus genome which were referred to as *Cit1,2RhaT* and *CitdGlcT-1*, *CitdGlcT-2*, and *CitdGlcT-3*, respectively [35]. These genes are not distributed evenly among different citrus species. Metabolite profiling by LC-MS in tobacco BY2 cells harboring the construct of *Cit1,2RhaT*, *CitdGlc-1*, or *CitdGlc-2* provided with different flavonoid substrates revealed that Cit1,2RhaT and CitdGlcT-1/-2 display high preference for flavanone-glucoside as substrate for glycosylation. Notably, RNAseq analysis in *C. maxima* fruit juice sac demonstrated that *CmdGlcT-2* is the most active *GlcT* and the dynamic pattern of *CmdGlcT-2* expression does not match with the availability of flavanone-7-O-glucoside during fruit development, consequently, there is high abundance of bitter naringin but no detectable bitterless flavanone-7-O-di-glucoside in pummelo. One of the possibilities to modify the bitterness of pummelo or citrus in general is through manipulation of the expression of *dGlcTs* to compete for the flavonoid substrates for better balance between flavor and nutrients [35]. Furthermore, the co-segregation of the presence of bitter-tasting naringin and *1,2RhaT* provides a genetic marker to assist citrus breeding for directed selection [35].

#### 2.2.4. CitLGTs

Limonoid is the other major type of bitter compound in addition to naringin in citrus fruit, including nomilin and limonin [8,24]. A study has shown that generally limonoids accumulate relatively high at young fruit or fruit expanding stage and then fall to very low in concentration or under detectable levels in mature fruit [36]. The majority of the bitter-tasting nomilin and limonin exists in fruit tissue as their non-bitter precursors nomilioate A-ring lactone (NARL) and limonoate A-ring lactone (LARL), respectively, before harvest or processing, and LARL is predominant [37]. The limonoid A-ring lactones in citrus are synthesized in leaves and transported into fruit where they are stored in the seeds and cell cytoplasm at a neutral to slightly alkaline pH environment [23,37,38]. During the juicing process, the precursors NARL and LARL are converted to the bitter nomilin and limonin, respectively, under the acidic condition (pH < 6.5) accelerated with the aid of limonoid D-ring lactone hydrolase (LLH) when the membranous sacs are ruptured [24,37]. The conversion of LARL to limonin can also occur naturally under acidic conditions when citrus fruits are young or when the fruits encounter freezing or mechanical damage [39,40]. The gradual conversion of the non-bitter precursors to the bitter limonoids is known as the process of delayed bitterness.

There are two forms of limonoid in citrus fruits and seeds, the bitter aglycones and non-bitter glucosides [40]. Limonoid aglycones are partly converted to the limonoid glucosides during fruit maturation [41], and soluble limonin glucoside is the predominant limonoid glucoside in citrus juice [42]. In citrus seeds, there exists the only natural repository of limonoid aglycones, where aglycone and glucoside of nomilin exceed those of limonin in concentration [43,44]. The conversion of aglycone of limonoids to glucoside of limonoids is catalyzed by limonoid glycosyltransferase (LGT) during citrus fruit maturation as natural debittering; specifically, LGT converts nomilin and LARL to nomilin glucoside and limonin glucoside, respectively [40].

Researchers have made progress on isolation and characterization of *LGT* genes from citrus species including satsuma mandarin (*C. unshiu*) and navel orange (*C. sinensis*) [39,45], Kinnow mandarin (*C. reticulata*) [40], sweet lime (*C. limettioides*), grapefruit (*C. paradisi*), and sour orange (*C. aurantium*) [26]. Overall these *LGTs* share high similarity in their coding sequences but they display different expression patterns during citrus development [26]. The satsuma mandarin and navel orange have two different alleles of *LGTs* where satsuma mandarin is heterozygous for *CitLGT-1*/*CitLGT-2* and navel orange is homozygous for *CitLGT-1*. The lack of delayed bitterness in satsuma mandarin was thought to be attributed to the presence of *CitLGT-2* which is highly expressed throughout the entire citrus fruit development, while the navel orange has the unpleasant delayed bitterness was on account of the much delayed expression (120 DAF) of *CitLGT-1* allele in the juice sac/segment epidermis [39]. Similar observations have been obtained from the citrus species with different levels of delayed bitterness demonstrating that the delayed presence of CitLGTs in the albedo tissue of citrus fruit is in agreement with the intensity of delayed bitterness [26]. The differential expression patterns provide a clue for marker-assisted selection during hybrid breeding and a target molecule for genetic engineering for debittering enhancement.

### 2.3. Sour

The sour taste of citrus fruits is mainly due to the presence of citric acid and malic acid, with citric acid as the dominant organic acid [46]. More specifically, the ratio of total soluble solid to titratable acid (TSS/TA) governs the quality and maturity of citrus fruits, which mainly reflects the ratio of sugar to citric acid content [47,48]. In the US, a minimum TSS/TA ratio of 7–9:1 is favored for oranges and mandarins, and 5–7:1 for grapefruit (http://www.yara.us/agriculture/). During citrus fruit ripening process, normally the sucrose content increases and the citric acid decreases, leading to the declined acidity and sourness; however, acidity increases with maturity in lemon causing the ratio of TSS/TA close to 1–2:1 [3,49]. High content of citric acid may mask the perception of sweet taste, influencing the overall taste of citrus fruit [50]. On the other hand, some natural citrus variants are acidless and thus taste sweet [51]. Citric acid is produced through tricarboxylic acid cycle (TCA cycles) in mitochondria and accumulated in vacuole of juice sacs during early-mid stage of fruit development and gradually degraded at the later stages in cytosol after released from vacuole [52,53]. Metabolic pathways were proposed to degrade citric acid to control its accumulation; its breakdown can be through the TCA cycle or γ-aminobutyric acid shunt (GABA shunt) in mitochondria, or through the acetyl-CoA pathway catalyzed by ATP-citrate lyase (ACL) in cytosol, characterized by the upregulated expression of related genes during fruit maturation [13,46]. It is also hypothesized that the citric acid after its export from vacuole to cytosol can be converted to 2-oxoglutarate catalyzed by the sequential action of cytosolic aconitase (Aco) and isocitrate dehydrogenase (IDH) for glutamate synthesis and further metabolism. The activity of aconitase was proposed to paly important role in the metabolism of citric acid during citrus fruit development [52,54], while a more thorough investigation is still needed. The metabolism network of citrate in citrus is summarized in Figure 4. Nevertheless, the vacuolar storage of citrate after export and degradation has a determinative effect on citrus fruit sour taste. Here we will give more details about understanding the underlying molecular mechanism regulating hyperacidity in vacuole and citrate flux into/out of vacuole.

#### 2.3.1. V-ATPase

It has been suggested that vacuolar H^+^-ATPase (V-ATPase) acts as the proton pump driving the H^+^ import into vacuole at the expense of ATP hydrolysis, and the resulting ΔpH and H^+^ electrochemical gradient facilitate the transport of citrate into vacuole [55,56]. Li et al. (2016) suggested that CitVHA-c4, one of the V-ATPases in *C. reticulata*, is involved in the citrate accumulation in vacuole considering its substantially high expression at fruit late development stage [57]. Biochemical assays indicated that the V-ATPase CitVHA-c4 can physically interact with the transcription factor CitERF13, and they can act alone or cooperatively in promoting citrus acid accumulation in transfected tobacco leaves [57]. An earlier research by Shimada et al. (2006) reported the isolation of a vacuolar H^+^-ATPase from *C. sinesis*, namely CsCit1, which functions as an active co-transporter of citrate and H^+^ in an electroneutral manner (citrateH^2−^/2H^+^) for citrate export from vacuole. The action manner of citrateH^2−^/2H^+^ possibly enhances the buffer capacity by selectively exporting the conjugate base of citrateH^2−^ thus favoring the maintenance of high vacuolar acidity during citrus fruit maturation [58]. These findings suggest the regulatory role of V-ATPase in vacuolar citrate buffer content and acidity.

#### 2.3.2. P-ATPase

The pH values in citrus such as sour lemon and lime can be as low as 2, which require ATPase with low H^+^/ATP ratio (<2) for hyper-acidification except for the regular V-ATPase for vacuolar lumens acidification [59,60]. In petunia, the interacting PH1 of P3B-ATPase (Mg^2+^ pump) and PH5 of P3A-ATPase (H^+^ pump) were reported to be the indispensable players for proton translocation across tonoplast against a larger electrochemical gradient for vacuolar hyperacidification in epidermal petal cells [61]. Strazzer et al. (2019) provided a wealth of evidence demonstrating that the two citrus homologs, CitPH1 and CitPH5, act as a heterodimer holding major responsibility for the hyperacidity in juice vesicles [62]. *CitPH1* and *CitPH5* are highly expressed in acidic (low pH) citrus fruit while strongly suppressed in non-acidic (high pH) one. Their downregulated expressions in high-pH citrus were attributed to the mutations in the transcription regulator CitAN1 (in sweet lemon and sweet oranges) or the reduced expression of the transcription factors CitAN1, CitPH3, and/or CitPH4 as a result of altered functions of their upstream trans-acting factors [62]. An in-depth investigation of those acidless variants in citrus species including citron, lemon, sweet lime, and sweet orange uncovered that a bHLH-type transcription factor, namely *Noemi*, is the potent contributor to citrus acidity except for its role in anthocyanin production [51]. The *Noemi* protein is a homolog of AN1 in petunia. The abolished or reduced expression of *Noemi* leading to acidless or reduced acidity was attributed to the mutation in *Noemi* alleles with deletion or transposon insertion in the coding region or small lesion adjacent to TATA-box in the promoter region [51]. The regulatory role of AN1 in *CitPH1* and *CitPH5* expression can account for the conserved regulatory role of *Noemi* in citrus acidity. Moreover, the CsPH4 in *C. sinensis* was characterized to be a physical interactor with *Noemi* and positively regulate the expression of *Noemi* as CsPH4-*Noemi* complex. Their expression levels positively correlated with the expression of *CsPH5* and the citrus acidity [63], providing another line of evidence in support of the citrus acidity regulatory complex involving CitPH1, CitPH5, and their transcription regulators. Additionally, Shi et al. (2019) further characterized CsPH8, another citrus homolog of petunia PH5, as a positive regulator of citrate content in citrus fruits based on previous profiling of *CsPH8* expression in citrus cultivars with different acidity [64,65]. These findings substantiate the notion that the steep pH gradient across the tonoplast is the powerful driving force for titratable citrate uptake into vacuole providing the sensation of sour taste [62,64]. CitPH1, CitPH5, CsPH8, and their transcription regulators provide promising genetic loci for molecular marker selection and genetic breeding in citrus for desired taste.

## 3. Citrus Genetic Modification Approach

Conventional citrus breeding is time-consuming and difficult as citrus has an extended juvenile period and possesses several special reproductive characteristics such as high heterozygosity, polyembryony, and self-incompatibility [66]. The parent lines with desirable traits are to be identified and genetically crossed, and it will take about another 20 years before the hybrid sets fruits for taste or flavor evaluation [9]. Ectopic expression of the citrus homolog of the *Flowering locus T* (*FT*) has been reported to be able to induce early flowering in trifoliate orange [67]. Dutt et al. (2005) also successfully generated early flowering Carrizo rootstock by constitutively expressing Clementine *FT* gene in the transgenic plant [68], and the induction of early flowering by FT was also documented in citrus plants of different genotypes by graft inoculation with the *Citrus leaf blotch virus*-based vector harboring the *AtFT* (Arabidopsis *FT* gene) or *CiFT* (Valencia late sweet orange *FT* gene) [69]. As documented, the transgenic plantlets can start to flower when they were 3–6 months old under the induction of the FT signal [68,69]. These findings provide a feasible approach to speeding up the breeding process. Given the mobile property of the FT signal, the exotic gene contamination can be avoided when the transgenic citrus is used as rootstock. The virus-based expression vector has this intrinsic advantage for transient expression of an exogenous gene of interest efficiently and systemically. It is encouraging to observe the precocious flowering phenotype induced by viral vector-medicated FT expression without compromising plant growth and citrus fruit development [69].

The molecular marker-assisted breeding can identify the citrus candidates with potential desired flavor at early stage [70]. Quantitative trait locus (QTL) mapping and other molecular marker genetic linkage analysis have been widely applied to explore certain associated target traits in marker-assisted selection program. A collection of single nucleotide polymorphisms based QTLs (SNP-based QTLs) was mapped to be associated with mandarin aroma volatiles production, which could facilitate the mandarin fruit flavor improvement [71]. The above-mentioned key genes controlling citrus sour and bitter taste qualities can serve as excellent DNA markers to evaluate the sensory traits of citrus fruit when the plants are still at juvenile stages. In addition, agrobacterium-mediated genetic transformation is a well-developed approach for direct introduction of gene of interest into citrus genome, and some advances have been made in using genetic element of plant origin as driving promoter or selection marker to avoid foreign gene introduction [72]. Other transformation methods such as poly ethylene glycol mediated (PEG-mediated) plasmid DNA uptake by protoplasts and biolistic DNA delivery also made success in citrus genetic modification [73,74]. Generally, antibiotic or herbicide resistance is used as selection marker for transgenic plantlets production. To overcome the risk of introduction of exotic DNA fragments into donor plants, it is worthwhile to mention that Ruby gene and citrus-derived embryo-specific promoter have been employed as visual reporter gene to monitor the success of transgene in citrus, as Ruby can act as a transcriptional activator of anthocyanin production [68,73]. The best-case scenarios would be upregulated or ectopic expression of the genes conferring “good” taste or repression of the genes controlling “bad” taste leads to enhancement of the overall taste in citrus. Moreover, the development of CRISPR/Cas system has facilitated the precision genome editing in plant, making it possible to generate transgene-free edited crops [75]. Several studies have reported using CRISPR/Cas9 system [76,77,78,79,80,81] or CRISPR/Cas12 system [82] as an efficient tool for genome editing in citrus. Great efforts have been focused on the improvement of citrus resistance to bacterial canker and HuangLongbing (HLB) through CRISPR/Cas system and conventional transformation strategies [83]. Interestingly, the isolation of monoembryonic early-flowering “Mini-Citrus”, its sequence homology to cultivated citrus species, and its amenability to CRISPR/Cas9 precise editing render it a potent model system to study gene functions in citrus [81]. By projected editing of the DNA sequence of the promoter or coding region of citrus taste-contributing genes to tune their expression, it would be achievable to improve the citrus fruit quality using CRISPR/Cas strategy in the long run.

## 4. Conclusions and Perspectives

This review outlines the major genes governing citrus taste properties including sweetness, bitterness, and sourness. The sweet sensation is largely determined by the sucrose content in citrus fruit, and the ratio of TSS/TA confers the dominance of sweet or sour taste of citrus. Sucrose is synthesized in leaves through photosynthesis and transported to citrus fruit, via a multiplayer-involved regulatory pathway. The content of titratable acid, mainly citric acid in the vacuole, delivers the sour taste and its import is greatly facilitated through the H^+^ gradient across the tonoplast. The characterized V-ATPase and P-ATPase and their regulators play essential roles in citrus acidification. The bitterness is influenced by the branch-forming glycosylation on flavanone-7-glucoside. The activity and availability of 1,2RhaT, 1,6RhaT, and dGlcT determine the bitter or non-bitter taste of citrus. The delayed bitterness is generated mostly by the conversion of non-bitter precursor to bitter limonin, and the glycosylation of limonoid catalyzed by limonoid glycosyltransferase (LGT) can act to debitter. The molecular evidence of these core taste-shaping genes by detecting their existence in citrus varieties would help evaluate the citrus fruit quality at juvenile stages. In addition, these genes can serve as a guide for genetic engineering for citrus taste modification.

The flavor of citrus fruit and juice is perceived and evaluated by our sense organs through their appearance (color and texture), aroma, and the juice and solid contents [84]. Other physical properties such as fruit size and shape, peel strength and thickness, and seedless also contribute to the customer acceptance of citrus fresh fruits [85]. The genetic makeup of flavor-forming pathways provides the basis for citrus flavor, while other favorable conditions such as temperature, fertilization and irrigation schedule, and harvest timing should be met as well for optimal citrus flavor and production. With the ever-growing knowledge about the underlying molecular mechanism shaping the citrus flavor and the advances in molecular genetic tools and techniques, the citrus breeding would be projected to grow faster than ever to provide consumers a variety of healthy and tasty citrus products.

## Figures and Tables

**Figure 1 ijms-20-06194-f001:**
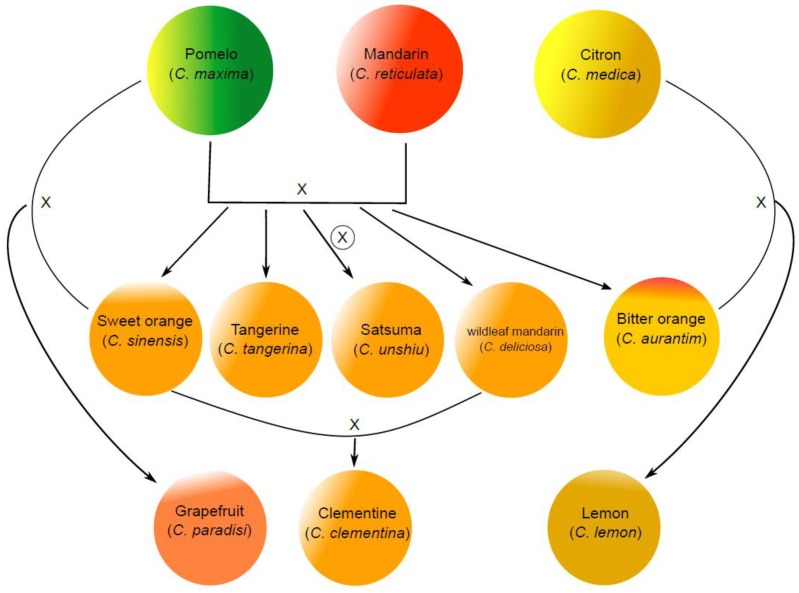
Proposed family tree of citrus species. The three ancestral citrus species of the main citrus species on the market include mandarin, pumelo, and citron. The commercial mandarins consist of tangerine, satsuma, and clementine, and they are hybrids between mandarin and pomelo. Satsuma is a highly inbred line from the hybrid of mandarin and pomelo. Clementine is the hybrid between sweet orange and willowleaf mandarin which are also the hybrid between mandarin and pomelo. Bitter orange is also a hybrid between mandarin and pomelo. Grapefruit is a hybrid between sweet orange and pomelo. Lemon is a hybrid between bitter orange and citron.

**Figure 2 ijms-20-06194-f002:**
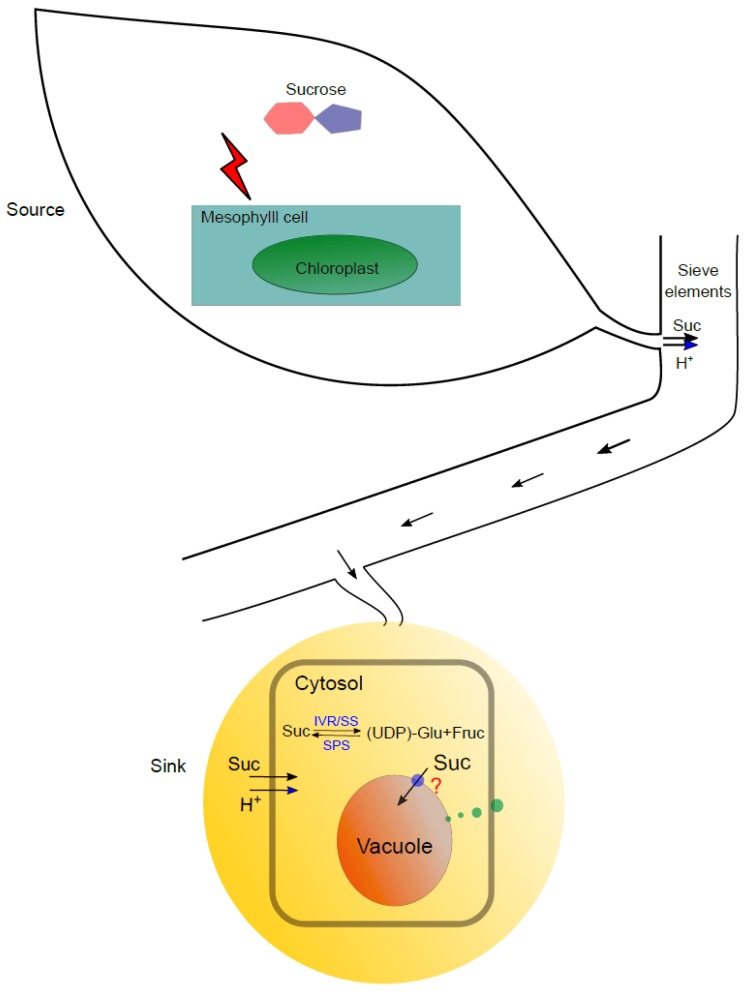
Schematic illustration of sucrose transport from source to sink in citrus. Sucrose is biosynthesized in leaf mesophyll cells through photosynthesis. The translocate sucrose is loaded to the sieve cells of phloem using H^+^ electrochemical potential gradient as driving force with the help of H^+^/sucrose symporter. Sucrose is transported in phloem following the turgor pressure in sieve elements towards sink tissue (citrus fruit) and unloaded by the symplastic or apoplastic pathway. Sucrose can be converted to fructose and glucose by IVR or fructose and UDP-glucose by SS, and it can be resynthesized through fructose and UDP-glucose by SPS in the cytosol. Sucrose uptake from apoplastic into cytosol is driven by the H^+^/sucrose symporter. The apoplastic sucrose can be directly incorporated into vacuole through endocytosis system, while the existence of an active transporter or H^+^/sucrose antiporter from cytosol to vacuole is still questioning. IVR—invertase; SS—sucrose synthase; SPS—sucrose-phosphate synthase.

**Figure 3 ijms-20-06194-f003:**
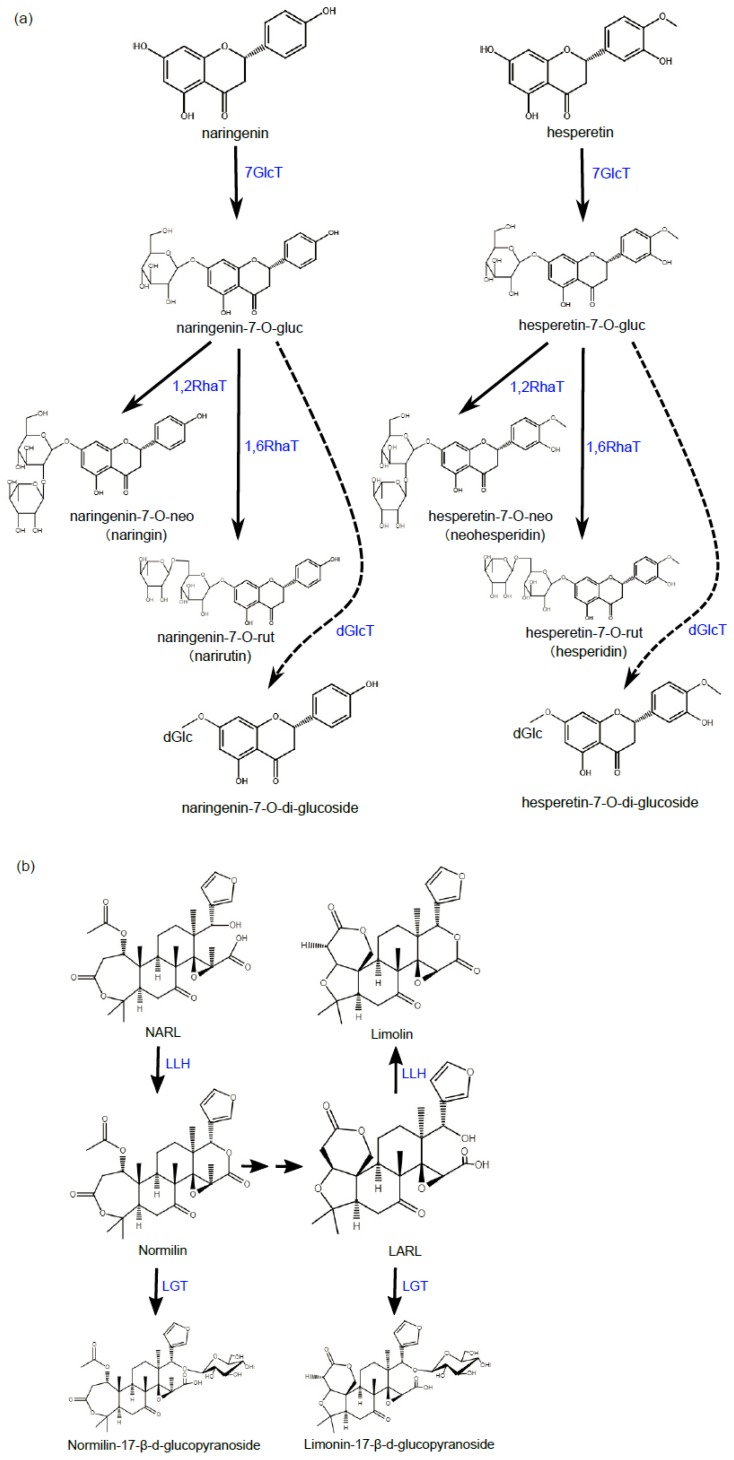
Flavanone glycosylation pathways, and limonoid aglycon or glycon formation pathways. (**a**) Two major flavanone compounds, naringenin and hesperitin, in citrus are exemplified. They can be converted to their glucoside derivatives at their C7 site by 7GlcT (7-glucose transferase), and further rahmnosylated on glucoside through C1,2 or C1,6 bond formation between rhamnose and glucose catalyzed by 1,2RhaT or 1,6RhaT, respectively. Consequently, the glycosylation of flavanone by neohesperidose (rhamylose-1,2-glucose) leading to the formation of naringin and neohesperidin confers bitterness and the glycosylation by rutinose (rhamylose-1,6-glucose) leading to the formation of narirutin and hesperidin confers non-bitterness. Flavanone-7-O-gluc can also be further glucosylated to form flavanone-7-O-di-glucocide (glucose-1,2-glucose as proposed) catalyzed by dGlcT; (**b**) Two major limonoid compounds, nomilin and limonin, in citrus are exemplified. Non-bitter NARL and LARL precursors can be converted to bitter-tasting nomilin and limonin, respectively, by LLH under acidic condition. Nomilin can be converted to LARL through multiple steps. Nomilin and LARL can also be converted to the non-bitter derivatives of nomilin-17-O-glucoside and limonin-17-O-glucoside, respectively, catalyzed by LGT. NARL—nomilioate A-ring lactone; LARL—limonoate A-ring lactone; LLH—limonoid D-ring lactone hydrolase; LGT—limonoid glycosyltransferase.

**Figure 4 ijms-20-06194-f004:**
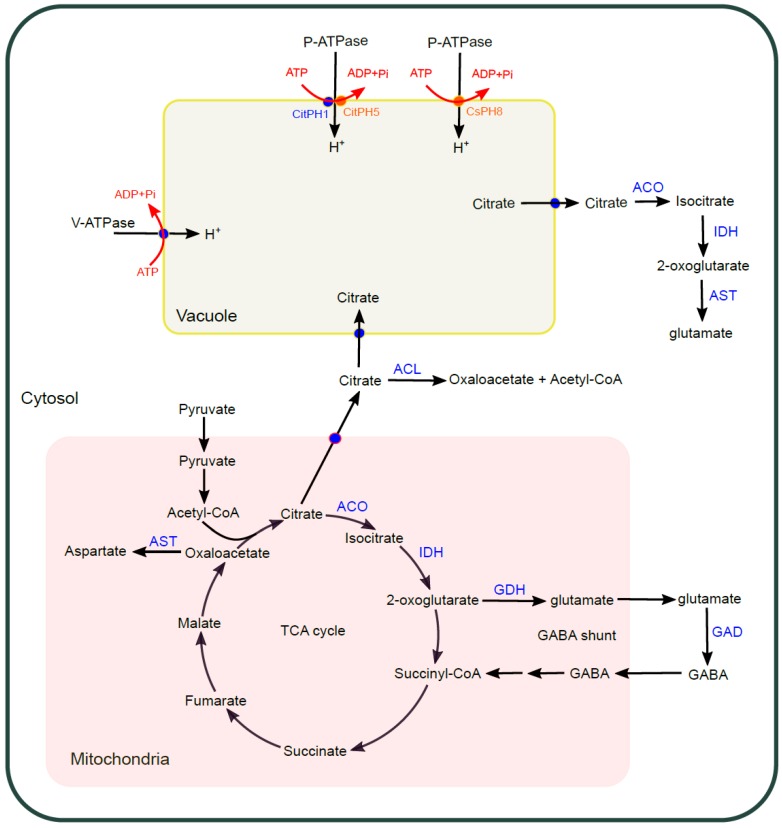
Citrate biosynthesis, transport, and accumulation in citrus. Citrate is synthesized in mitochondria through TCA cycle. In mitochondria, citrate can be degraded through TCA cycle initiated by the enzymatic actions of mitochondrial ACO and IDH or through GABA shunt characterized by the functioning of enzymes of GDH and GAD with the shuttle of glutamate and GABA between mitochondria and cytosol. Citrate can be exported from mitochondria to cytosol and further degraded by the action of ACL. Citrate exported from mitochondria to cytosol can be imported to vacuole with the help of H^+^ electrochemical gradient and pH gradient conferred by the actions of V-ATPase and P-ATPase. Vacuolar citrate can be exported to cytosol by transporters and be degraded by the action of cytosolic ACO and IDH. The integral metabolic pathways affect the citrate content in vacuole and sour taste of citrus fruit. ACO—aconitase; IDH—isocitrate dehydrogenase; GDH—glutamate dehydrogenase; ACL—ATP citrate lyase; GABA—gamma-aminobutyric acid; AST—aspartate aminotransferase.

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
