# Peer review of "Citrus Taste Modification Potentials by Genetic Engineering"

_ijms, 2019, doi:10.3390/ijms20246194_

Round 1

Reviewer 1 Report

This MS is O.K. but not outstanding; it indicates the potential for genetic engineering but, in my opinion, is not specific enough. Perhaps it is still too early  - have any real genetic engineering achievements been reported?

Now something else. A recent line of research must be included; one representative paper attached.

Author Response

Response to comments from reviewer 1

(1) This MS is O.K. but not outstanding; it indicates the potential for genetic engineering but, in my opinion, is not specific enough. Perhaps it is still too early  - have any real genetic engineering achievements been reported?

Answer: Thank you for this comment that is to the point. We were trying to outline some key genes involved in shaping citrus taste as a reference for molecular marker-assisted selection and genetic engineering in the future. We agree that we could not present some evidence specific enough to convince that the genetic manipulations on those genes would definitely enhance citrus taste, as there is no publicly available publication regarding the genetic engineering achievements in citrus taste modification so far to the best of our knowledge.

(2) Now something else. A recent line of research must be included; one representative paper attached.

Answer: Thank you for this constructive suggestion to improve our manuscript. We have integrated the two recent studies on Noemi (Butelli et al., 2019 and Zhang et al., 2019) in our manuscript as shown in line 332-343.

Reviewer 2 Report

Paper gives a good overview on the citrus taste and its molecular background. The genetic/breeding section is less detailed than expected by the title. It is recommended to go into more detail in section "3. Citrus genetic modification approach".

English: It is recommended to shorten sentences length. This would make the paragraphs easier to read.

Author Response

Response to comments from reviewer 2

(1) Paper gives a good overview on the citrus taste and its molecular background. The genetic/breeding section is less detailed than expected by the title. It is recommended to go into more detail in section "3. Citrus genetic modification approach".

Answer: Thank you for the positive comments on our manuscript and the suggestion to further improve our manuscript. We have added more details regarding molecule-assisted selection and genetic engineering approaches which have been successfully performed in citrus in the section of citrus genetic modification approach.

(2) English: It is recommended to shorten sentences length. This would make the paragraphs easier to read. 

Answer: Thank you for pointing out this critical point to improve the quality of our manuscript. We have rephrased some sentences to make them more readable (Line 105-106; line 315-317; line 366-367). In addition, we have corrected some typos and grammatical errors in our manuscript.

Round 2

Reviewer 1 Report

It has certainly been improved as recommended. I therefore approve its publication in IJMS

Reviewer 2 Report

Thank you a lot for considering the corrections of all reviewers, which improved the work.